# Immunotherapy Strategies for Gastrointestinal Stromal Tumor

**DOI:** 10.3390/cancers13143525

**Published:** 2021-07-14

**Authors:** Junaid Arshad, Philippos A. Costa, Priscila Barreto-Coelho, Brianna Nicole Valdes, Jonathan C. Trent

**Affiliations:** 1Hematology-Oncology Department, Sylvester Comprehensive Cancer Center, University of Miami, Miami, FL 33136, USA; dr.junaidarsh@gmail.com; 2Internal Medicine Department, Jackson Memorial Hospital, University of Miami, Miami, FL 33136, USA; philippos.costa@jhsmiami.org (P.A.C.); priscila.barretocoe@jhsmiami.org (P.B.-C.); 3Miller School of Medicine, University of Miami, Miami, FL 33146, USA; briannanicole22@med.miami.edu

**Keywords:** gastrointestinal stromal tumors, GIST, immunotherapy, checkpoint inhibitors, antibodies, lymphocytes, cytokine

## Abstract

**Simple Summary:**

Gastrointestinal stromal tumors (GIST) are the most common type of gastrointestinal sarcomas. Recent treatment advancements have led to a significant improvement in the survival of patients living with GIST. Nevertheless, patients with metastatic GIST will eventually require multiple lines of treatment. The use of immunotherapies could potentially fill the role of such treatments and potentiate the current therapies. This article reviews the mechanism by which the immune system interacts with GIST and discusses several studies on the use of immunotherapies in GIST.

**Abstract:**

Gastrointestinal stromal tumors (GIST) are the most common mesenchymal soft tissue sarcoma of the gastrointestinal tract. The management of locally advanced or metastatic unresectable GIST involves detecting KIT, PDGFR, or other molecular alterations targeted by imatinib and other tyrosine kinase inhibitors. The role of immunotherapy in soft tissue sarcomas is growing fast due to multiple clinical and pre-clinical studies with no current standard of care. The potential therapies include cytokine-based therapy, immune checkpoint inhibitors, anti-KIT monoclonal antibodies, bi-specific monoclonal antibodies, and cell-based therapies. Here we provide a comprehensive review of the immunotherapeutic strategies for GIST.

## 1. Introduction

Gastrointestinal stromal tumor (GIST) is the most common mesenchymal neoplasm of the gastrointestinal (GI) tract, accounting for 80% of all GI sarcomas and 0.1 to 3% of all GI malignancies [1], affecting 1.1 every 100,000 persons per year worldwide [2]. While GIST can occur throughout the GI tract, 60% of the cases present in the stomach and around 30% in the small intestine [3,4]. GIST can also occur at any age but are most common in the 40s–60s age group with similar male to female incidence [3,4]. While it is known that GIST arises from the same lineage as the interstitial cells of Cajal (ICC), which function as pacemaker cells and regulate the GI tract peristalsis, it is not clear if these tumors arise directly from the mature cells or their precursors. The cell-surface tyrosine kinase KIT receptor is vital to the function and survival of ICCs, with 85% of GIST occurring due to mutually exclusive activating mutations of KIT (CD-117) or platelet-derived growth factor receptor alpha (PDGFRA) [5,6].

The treatment landscape of advanced and metastatic GIST involves the identification of molecular alterations and the availability of targeted agents. Almost 70–75% of GIST cases have KIT mutations. About 10% of GIST with no KIT mutations are considered KIT wild type and have PDGFR, BRAFV600E, N-RAS, PIK3, N-TRK mutations, or are SDH deficient via deletion or epigenetic silencing [7,8]. The Food and Drug Administration (FDA) has approved imatinib, sunitinib, regorafenib, and ripretinib for the treatment of advanced or metastatic GIST in the first-, second-, third- and fourth-line setting, respectively. Imatinib mesylate, a KIT tyrosine kinase inhibitor, is the first-line systemic therapy for GIST. It can achieve a clinical response in 80% of patients leading to an impressive median overall survival (OS) of 3.9 years and progression-free survival (PFS) of 1.9 years [7,9]. Despite the excellent response, it is not curative as secondary KIT mutations frequently develop over time, resulting in acquired resistance. Additionally, about 14% of GIST patients present with de novo or primary resistance to imatinib [10,11].

There has been ongoing work to discover new targets, especially in the growing era of immunotherapy. The microenvironment of GIST is populated by tumor-infiltrating immune cells. These cells have an essential role in tumor surveillance and can potentially remove imatinib resistant clones, enhancing imatinib activity. Here we review the role of immunotherapy in GIST treatment and the ongoing trials [12].

## 2. Disease Biology

### 2.1. Principles of Tumor Immunology

The immune system is comprised of different types of bioactive molecules, cytokines, proteins, and cells that collectively recognize and defend against foreign proteins or antigens. To provide protection and maintain the state of homeostasis, the immune system relies on two forms of immune responses: innate and adaptive.

The innate response is the first line of defense, being time-independent and nonspecific. Its cellular components include mast cells, dendritic cells, macrophages, neutrophils, and NK cells. Also included in the innate response are the interferon alfa and gamma. Interferon alfa stimulates immune cells to produce interferon-gamma. Interferon-gamma then activates the NK cells and CD8 cytotoxic cells, which are responsible for immune surveillance. On the other hand, indoleamine-2,3-dioxygenase (IDO) suppresses NK and CD8 cytotoxic T cells and activates regulatory T cells, decreasing the immune surveillance. The adaptive response is the second line of defense, being time-dependent and specific to different antigenic stimuli. Its cellular components include B cells (with an emerging role in antitumor immunity) and T cells. This response’s specificity is achieved through B cell antibody production, T cell stimulation, and memory functions [13].

A balance between stimulatory and inhibitory interactions always exists in the immune system. A shift in this balance can lead to the eradication of tumors or tumor escape from the immune system leading to disease progression [14]. The identification of inhibitory signals, such as the ligand to PD-1 (PD-L1), led to a new class of immunotherapeutic drugs that hinders immune effector inhibition, which may reinvigorate pre-existing anti-cancer immune responses [15,16]. Other vital parts of the immune response are immunomodulation, with a complex interaction of positive and negative regulatory signals, tumor-specific mutations, stromal-matrix supportive function, immune evasion, and suppression, as well as immune recognition, related to the mutational burden in cancer and its potential impact on patient outcomes [15].

### 2.2. GIST Cellular Immunology

Tumor-infiltrating immune cells populate the microenvironment of GIST. The most common cells infiltrating GIST are tumor-associated macrophages (TAMs), both type 1 macrophages (M1) and the alternatively activated type 2 macrophages (M2), followed by CD3^+^ T lymphocytes [17]. Less frequent immune cells are tumor-infiltrating neutrophils, dendritic cells (DCs), natural killer cells (NK), natural killer T cells (NKT), gamma delta T cells, and B cells [6].

Regulatory T cells (T-regs) are detectable in most GISTs, and it correlates significantly with the quantity of M2 macrophages. This suggests that local anti-inflammatory M2 macrophages stimulate the influx of T-regs. This finding, together with the decreased ratio of CD8^+^/T cells and the reduced HLA class 1 molecules (which might prevent tumor cell detection by lymphocytes), implicates that GIST microenvironment is immune suppressive [17].

Tumor-infiltrating lymphocytes (TILs) in GIST are constituted predominantly of CD3^+^ T cells, including CD8^+^ cytotoxic T lymphocytes (CTLs), CD4^+^ T helper type 1 lymphocytes (Th1), CD4^+^ T helper type 2 lymphocytes (Th2), Forkhead box P3 protein (FoxP3) expressing T-regulatory cells (T-regs,) and IL-17^+^ T helper cells (Th17). CD3^+^ T cells are more common in metastases, in tumors originated in the small intestine or colon, and in tumors with a high proliferation index (PI) [18]. Additionally, the number of CD3^+^ T cells correlates with smaller tumor size, decreased relapse rate, and higher progression-free survival [18,19]. On the other hand, NK cells are less frequent than CD3^+^ T cells in GIST and are associated significantly with lower PI, stomach localization, reduced relapse rate, and improved progression-free survival [19]. Additionally, in patients treated with adjuvant imatinib, NK, and CD3, cell infiltrates are associated with a reduced relapse rate [19].

New studies have described the predictive signatures such as the expanded IFN-γ-induced immune signature (EIIS) and the T-cell-inflamed signature (TIS) within the GIST microenvironment showing the abundance of CD4^+^, CD8^+^ T cells, and M2 macrophages. EIIS shows positive correlation with PDL1 expression, which in turn is positively correlated to CD8A and CD8B gene expression. TIS also correlates well with the response to immune checkpoint inhibitors in GIST in the form of TIS score [20]. New studies have looked at the association of these immune signatures with targeted mutations of KIT/PDGFR. PDGFRA mutant GIST has a stronger immune signature profile with an abundance of CD8^+^ and CD45^+^ cells, making it a more favorable molecular environment for cytolytic activity as compared to KIT mutant GIST [21]. PDGFR D842V mutant GIST has the highest immune signature profile with an increased TIS score, CD8^+^ lymphocytes, and M2 macrophages, compared to KIT and non D842V mutation GIST [22]. D842V Mutant GIST is also associated with highest number of neo-epitopes, as many as six, as compared to other KIT and non D842V mutant GIST [22]. The correlation of the presence of the neo-epitopes with immune response remains to be elucidated.

### 2.3. Imatinib and the Immune System

Imatinib has revolutionized the GIST treatment, and its antitumor activity is enhanced by its interaction with the immune system [3]. Balachandran et al. described an animal model in which imatinib increased the frequency, activation, and proliferation of intra-tumor CD8^+^ T cells and apoptosis of T-reg cells, producing an increase in the ratio of CD8/T-reg, but not changing the amount of IL-4, IL-17, or interferon-gamma (NF-γ) produced by CD4^+^ T cells or the percentage of myeloid cells, B cells, NK cells or NKT cells. Nevertheless, there is an increase in the production of interferon-gamma by NK cells [23]. Also, imatinib therapy reduces IDO expression, leading to activation of CD8^+^ T cells and apoptosis of tumor-infiltrating T-reg cells [23,24].

The efficacy of imatinib is also mediated by its influence on NK cells. It inhibits KIT receptors in dendritic cells, promoting crosstalk between DC and NK, and the production of γINF. Delahaye et al. report that this imatinib effect can be related to better GIST patient outcomes. Imatinib responders had higher levels of NK cells and INF-gamma after 2 months of treatment compared to pre-treatment levels. Such effect was not seen in non-responders [25]. Additionally, post-imatinib-treated tumors show an increased amount of infiltrating NK cells, while treatment with imatinib significantly decreases the T-reg cell population [19]. With these findings, the accumulation of T-reg cells together with the loss of NK cells can be considered predictors of poor imatinib response.

## 3. Immunotherapies in GIST

The role of immunotherapy in sarcomas is growing fast, with some impressive responses being described [26]. The preservation of the proper maturation of the leukocytes makes immunotherapy an attractive adjuvant option in GIST treatment since imatinib, except in rare cases, does not lead to leukopenia [27]. Considering that immunotherapy can have a role in GIST, several pre-clinical and clinical trials have been conducted (summarized in Table 1). Among these immunotherapeutic possibilities, we have cytokine-based therapies, immune checkpoint inhibitors, anti-KIT antibodies, and cellular therapies (summarized in Figure 1).

### 3.1. Cytokine-Based Therapy

Cytokine therapy can switch the balance of the immune system in favor of increasing immune surveillance. The immune surveillance would eradicate drug-resistant clones in GIST, leading to a better imatinib response. This increase in immune surveillance is achieved by promoting a Th1 response [6]. To test this notion, a clinical trial combining peg-interferon alfa-2b with imatinib was conducted to treat stage III/IV GIST [23].

In the pre-clinical phase, the trial authors converted an anergic T-lymphocyte from a synovial sarcoma patient into an active CD8 with antitumor activity in vitro [23]. This was achieved by presenting an antigen cocktail to DCs, that produced IL12 (inducer of a Th1 response in vivo), stimulating the CD8 cells [23]. After activation, these CD8 cells strongly targeted tumor antigens [23]. The results of this experiment proved that cytokine therapy can transform immune tolerant cells into antitumor cells.

In the trial’s clinical phase, eight treatment naïve patients were enrolled and received the combination of imatinib and peg-interferon alfa-2b. To assess if interferon alfa was able to promote a Th1 response, pre-treatment and post-treatment peripheral blood were analyzed [23]. A significant increase in the number of CD4, CD8, and NK cells was noted in the post-treatment blood, associated with an increased interferon-gamma production [23], the key element of a Th1 response. On top of that, in comparison to pre-treatment histology, post-treatment tumors had a substantial increase in the number of infiltrating lymphocytes, that strongly stained for interferon-gamma [23]. This was evidence that peg-interferon alfa-2b was able to promote a Th1 response.

Regarding outcomes, all eight patients had partial responses, with one of the patients having a complete pathological response [23]. Out of eight patients, only two had PD disease during the study’s follow-up (median follow-up of 3.4 years). The patient with a CR had no PD during his 3.6 years follow-up [23]. The drug was overall well tolerated. One patient had grade 3 neutropenia, requiring dose adjustment of the interferon, and another had a skin rash requiring corticosteroids [23]. All patients experienced transient low-grade fevers and flu-like symptoms that are expected with such treatment [23].

### 3.2. Immune Checkpoint Inhibitors

Checkpoint inhibitors work by preventing checkpoint proteins from binding with their related proteins. This leads to an activation of the immune system, increasing its capacity to eliminate cancer cells. Candidate targets for this approach include programmed cell death protein-1 (PD1) and its ligands (PDL1/PDL2), CTL-associated antigen 4 (CTLA-4), T cell immunoglobulin, and mucin protein 3 (Tim3), and its ligand galectin-9 [24].

Immune cells express PD1, whereas different tissues, including tumors, express PDL1. The interaction between PD1 and PDL1 leads to immune evasion of cancer cells. Although PDL1 has an immunosuppressive function, it was noted that an increased expression of PDL1 in GIST cells carries a favorable prognosis [28]. This counterintuitive finding could be explained by the fact that a strong immune response against the tumor cells might increase the expression of PDL1 by selective pressure [28]. The ability to create an immunosuppressive microenvironment by the tumor is further evidenced by the fact that CD4 lymphocytes infiltrating GIST have a higher expression of PD1 in comparison to lymphocytes from matched blood [29]. Blocking the PD1-PDL1 pathway decreases the probability of the tumor bypassing the immune system. This was evident in mice with GIST, where the PD-1 and PD-L1 blockade potentiated imatinib’s effects [29]. A phase 1b/2 trial to assess the safety and efficacy of an anti-PD1 antibody named spartalizumab (PDR001) was started in 2018. Patients with metastatic or unresectable GIST with prior failure of imatinib, sunitinib, and regorafenib are being recruited, with the trial’s conclusion in August 2020 [30].

The combination of CTLA-4 blockade with imatinib is a conceivable therapeutic option. In mice with GIST, a specific antibody that can blockade CTLA-4 associated with imatinib led to a superior response than each treatment alone [24]. The investigators also noticed an increase in interferon-gamma production by CD8 T cells during the combination treatment [24]. The increase in interferon-gamma production intensified the Th1 response and could explain the synergic response seen. Based on the synergic effect shown by the pre-clinical data, investigators conducted a phase 1b trial of dasatinib (a TKI) plus ipilimumab (a CTLA-4 antibody) [31]. In the trial, 20 patients with GIST were enrolled. The most common adverse effect was anemia (94%), also being the most common grade 3 adverse effect (21%) [31]. Of the 13 evaluable patients, 7 (54%) had a partial response by Choi criteria, with the median duration of the response being 82 days. The trial’s reported progression-free survival was 2.8 months, which is similar to that reported in the literature [31]. In that trial, one patient with suppressed indoleamine-2,3- dioxygenase 1 (IDO1) had stable disease for 4.75 months, while two patients that did not have IDO1 suppression had progressive disease [31]. The authors hypothesized that IDO1 inhibited the T cells, and ipilimumab could not overcome the negative regulation. Based on the idea that IDO1 can negatively regulate the effects of PD-1 inhibitors, a trial combining an anti-PD-1 and an IDO1 inhibitor was designed and started in 2017. Patients with GIST that failed at least two TKI regiments will receive epacadostat, an IDO1 inhibitor, and pembrolizumab, a PD-1 inhibitor [32]. The endpoint of the trial is overall response rates. It is estimated to enroll 23 participants and will be concluded in December 2021 [32].

A randomized phase 2 study with nivolumab (anti CTLA 4) versus nivolumab plus ipilimumab (anti PD1) for patients with metastatic or unresectable GIST was started in 2016 [33]. In this trial, 40 patients were randomized 1:1 in two different groups: Nivolumab 240 mg every 2 weeks for 2 years or until disease progression, or nivolumab 240 mg every 2 weeks with ipilimumab 1 mg/kg every 6 weeks for 2 years or until progression. The trial’s preliminary results are promising, with 31% of patients in the nivolumab arm and 23% in the combination arm achieving more than 6 months of progression-free survival. In the preliminary publication of the results, in the nivolumab only arm, 7 out of 15 patients had SD, and in the nivolumab plus ipilimumab, 2 out of 12 had SD and 1 had a PR. Interestingly, the patient with a PR had a 70% shrinkage in this tumor [33]. Overall, these drugs have been well-tolerated, with 4 patients in each arm having a grade 3 or 4 adverse effect [33].

T-cell immunoglobulin mucin 3 (TIM3) is expressed in immune cells, while galectin-9 is the ligand expressed in tumor cells. The TIM3/galectin-9 pathway is currently under investigation for immune checkpoint therapies. In CD8 cells infiltrating GIST, TIM3 is upregulated, while galectin-9 is expressed on 2/3 of GISTs [6]. The presence of both in GIST makes this pathway an interesting target for developing checkpoint inhibitor drugs.

### 3.3. Anti-KIT Antibodies

Monoclonal antibodies can also enhance immune cell-mediated tumor clearance. SR1 is an anti-KIT monoclonal antibody that can slow the growth of human GIST cells in vitro and GIST mice [34]. After treatment with the antibody, tumor cells showed a decrease in KIT expression, leading the authors to consider that the mechanism by which the antibodies affect the cells is a downregulation of KIT, and through augmenting phagocytosis of GIST cells by macrophages [34]. It was also found that imatinib resistant and sensitive cells were equally inhibited by the antibody [34], suggesting that this treatment strategy could be used in adjunction to imatinib. Additionally, the antibody increased the phagocytosis of GIST cells [34], indicating that the antibody can increase tumor cells’ clearance by the immune system.

An anti-KIT antibody conjugated to a microtubule destabilizing maytansinoid was also tested in vitro and in monkeys [35]. This antibody conjugated drug, LOP628, had a strong anti-proliferative effect on c-KIT cells in vitro and was well-tolerated in monkeys. Interestingly, just the antibody, not conjugated to the drug, had little to no effect on the cells [35]. The effect of the antibody conjugated drug was superior in imatinib-resistant GIST cells [35], being also a promising therapeutic option.

### 3.4. Bi-Specific Monoclonal Antibodies

Somatostatin is a natural growth hormone inhibitory neuropeptide with anti-secretory action. It inhibits cell proliferation in both normal and tumor cells by binding into trans-membrane G-protein receptors named somatostatin receptors (STTRs). Its anti-proliferative function is due to inhibitory effects on MAPK and PI3K pathways [36]. Octreotide, a somatostatin analog, demonstrated anti-proliferative function in advanced intestinal neuroendocrine tumors [36]. GIST tumors highly express STTRs, with SSTR2 being present in 87% of GIST samples [36]. It was also found that tumors that highly express SSTR have a worse prognosis [36]. A bispecific antibody was designed to bind CD3 and SSTR2 with the goal of triggering a potent cytotoxic T-lymphocyte response. Currently, a phase 1 trial enrolling patients with neuroendocrine tumor (NET) and GIST is ongoing to assess the drug’s tolerability [37].

### 3.5. Cellular Therapy

Genetically engineered CD8 T cells generated with specificity toward GIST cell antigens can potentially be used in the treatment of GIST patients. A chimeric antigen receptor (CAR) T cell was produced with anti-KIT activity. Anti-KIT CAR T cells were able to bind GIST cells, produce interferon-gamma, and lyse the cells in vitro [38]. This antitumor effect was also seen in imatinib resistant cells [38]. In mice with GIST, the anti-KIT CAR T cells’ infusion resulted in a significant reduction in tumor growth [38]. The antigen release by the lysed cells, associated with an increase in the inflammatory response, may stimulate the endogen tumor surveillance, and also contribute to tumor regression [38].

## 4. Discussion

Immunotherapies are potential therapeutic agents for GIST. They are currently not the standard of care, but the clinical trials are encouraging. Meaningful responses were seen with the use of cytokine therapy [23] and ICI [31], and pre-clinical data are bringing a new horizon of treatments into view [34,35,37,38]. Although there are exciting advances in immunotherapies in GIST, these advances still lag as compared to other sarcomas and solid tumors such as melanoma and non-small cell lung carcinoma [26,39]. This is not due to the ineffectiveness of immunotherapies in GIST but to the high efficacy of TKIs, which naturally draw the attention of researchers. Nevertheless, despite TKIs, all metastatic GIST will invariably progress, leaving an unmet need for further therapies.

The role of immunotherapies in the treatment paradigm of GIST is still unclear. At the current stage of development, they would most likely be used in synergy with TKIs, as seen in the existing clinical trials [23,31]. However, this will have to be weighed against increased toxicity and financial burden. In the future, as new therapies develop, the role of immunotherapies could be expanded to GIST patients refractory to standard treatments or to a subset of patients harboring specific tumor characteristics after evaluation by next-generation sequencing or circulating tumor DNA [40].

## 5. Conclusions

Immunotherapy holds therapeutic potential for all types of sarcomas, including GIST. While immunotherapies are in an early stage of development for GIST treatment, it has been shown that they can potentiate traditional therapies such as KIT-TKIs. Future strategies should include combinations of immunotherapies and KIT-TKIs, for example, interferon alfa, ICI and a KIT-TKI.

## Figures and Tables

**Figure 1 cancers-13-03525-f001:**
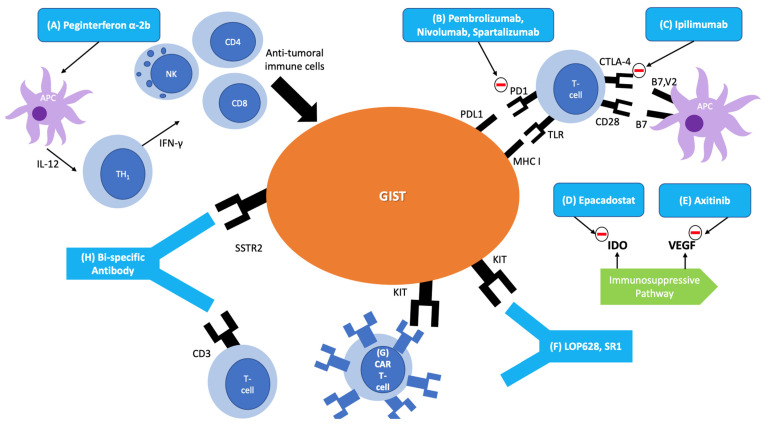
Immunotherapies in GIST. (**A**) Cytokine Therapy: Peginterferon alpha-2b is a cytokine-based therapy that is used in combination with imatinib to treat stage III/IV GIST. It functions by presenting an antigen cocktail to an antigen presenting cell (APC) to induce an IL-12 mediated TH1 immune response. The TH1 response secretes interferon-gamma and activates anti-tumoral immune cells. (**B**–**E**) Immune Checkpoint Inhibitors: (**B**) Pembrolizumab, Nivolumab, and Spartalizumab are programmed cell death protein-1 (PD1) inhibitors, preventing the PD1 and PDL1 interaction that leads to immune suppression and evasion of the immune system by cancer cells. (**C**) Ipilimumab is a CTLA-4 antibody, inhibiting the receptor’s function of sending inhibitory signals to T cells. (**D**) Epacodostat inhibits indoleamine-2,3-dioxygenase (IDO), an enzyme that decreases immune surveillance by downregulating NK and CD8 cells while upregulating T-regs. (**E**) Axitinib inhibits *VEGF*, thereby inhibiting the immunosuppressive function of *VEGF*. (**F**) Anti-KIT Antibodies: microtubule destabilizing maytansinoid (LOP628) and SR1 are monoclonal antibodies able to slow the growth of GIST via downregulation of KIT. (**G**) Cellular Therapy: Anti-KIT chimeric antigen receptor (CAR) T cells bind to GIST tumor cells leading to cell lysis. (**H**) Bi-specific monoclonal antibodies: binds to a somatostatin receptor (SSTR2) on tumor cells and CD3 to trigger a potent cytotoxic T cell response.

**Table 1 cancers-13-03525-t001:** Clinical trials on immunotherapy in GIST.

Trial	Phase	Drug	GIST Population	Results
NCT00585221	Phase 2	Peginterferon alfa-2b plus Imatinib	Stage III/IV	7 PR and 1 CR (100%). 2 (25%) patients progressed during the 3.6 m follow-up.
NCT01643278	Phase 1	Dasatinib plus Ipilimumab	Advanced/unresectable refractory to imatinib and sunitinib	7 PR (53%), 3 SD (23%), and 3 PD (23%), with a median duration of response of 82 days
NCT02636725	Phase 2	Axitinib plus Pembrolizumab	Unresectable and refractory to first-line therapy	No response seen in GIST patients (*n* = 3)
NCT03609424	Phase 1b/2	Spartalizumab (PDR001)	Metastatic or unresectable	Ongoing
NCT03411915	Phase 1	XmAb^®^18087	Advanced or metastatic refractory to all FDA-approved therapies	Ongoing
NCT03291054	Phase 2	Epacadostat plus Pembrolizumab	Unresectable or metastatic refractory to imatinib and another TKI	Ongoing
NCT02880020	Phase 2	Nivolumab Monotherapy versus Nivolumab Combined with Ipilimumab	Metastatic or Unresectable	Ongoing
NCT03475953	Phase 1/2	Regorafenib with Avelumab	Metastatic or Unresectable	Ongoing
NCT02406781	Phase 2	Pembrolizumab with metronomic cyclophosphamide	Metastatic or Unresectable after imatinib and sunitinib.	Ongoing
NCT04258956	Phase 2	Avelumab with Axitinib	KIT or PDGFRA positive advanced or metastatic, with no more than 3 previous therapies, which must include imatinib and sunitinib	Ongoing
NCT02834013	Phase 2	Nivolumab and Ipilimumab	Progressed in at least one line, and no further approved standard therapy that prolongs survival	Ongoing

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
