# Peer review of "Immunotherapy Strategies for Gastrointestinal Stromal Tumor"

_cancers, 2021, doi:10.3390/cancers13143525_

Round 1

Reviewer 1 Report

This manuscript (Manuscript ID: cancers-1246266, discussing the immunotherapy strategies for gastrointestinal stromal tumor) provides a clear overview of targets of immunotherapies for potential future use in GIST. Since discovery of oncogenic mutations in the KIT and PDGFRA tyrosine receptor, GIST became a model for development of molecular targeted therapy. Various TKI’s led to impressive improvement of clinical outcomes in advanced GIST, when compared to era before TKI’s were available. Immunotherapies are in an early stage of development for GIST and might hold therapeutical potential. However, its value compared to traditional TKIS’s and TKI’s in development, should be awaited. 

Introduction

Please include in the first the main GIST epidemiological; the incidence, the primary sites of occurrence, age, and gender distribution, for the reader to have a better context of the disease.

Lines (43-44): please specify clinical benefit (progression free survival, overall survival) to emphasize the dramatically improved prognosis of patients with advanced GIST after treatment with imatinib and eventually to provide the reader a reference for future novel therapies.

2.2 GIST Cellular Immunology

Paragraph 2: unclear purpose of this paragraph. Please elaborate. 

Paragraph 3 (lines 93-94): please provide reference for statement on decreased ration of CD8+/T cells and reduced HLA class 1 in GIST.

Line 102: please define the meaning of acronym PI.

  1. Immunotherapies in GIST

Line 131-132: although not common, neutropenia (and rarely agranulocytosis) induced by imatinib, has been observed in GIST patients. Please mention this observation with reference (case report, PMID: 25810235). 

Table1.

Please check  the list of relevant trials for completeness. 

Consider to add following trials with immunotherapy in GIST in table 1.

Regorafenib with Avelumab (NCT03475953)

Pembrolizumab in combination with cyclophosphamide in advanced sarcomas (NCT02406781)

Axitinib in combination with avelumab (NCT04258956)

3.1. Cytokine-Based Therapy

Please mention whether the patient, in the clinical trial (imatinib + peg-intreferon alfa-2b) were pre-treated with imatinib or any other TKI’s. And what was duration of response?

3.2. Immune Checkpoint Inhibitors

Lines 239-244: Only 3 patients (of 33 patients) were GIST patients in this trial (VEGF inhibitor + pembrolizumab). If the goal of the review is to include all the studies performed with immunotherapy in GIST patients, more studies should be mentioned for completeness (including trials with only few GIST patient). However, in my opinion only relevant trials with adequate numbers of GIST patients should be described. Therefore describing of this trial could be skipped.

3.3 Anti-KIT antibodies

It’s doubtful whether therapy with anti-KIT antibodies can be defined as immunotherapy. Consider to delete this paragraph.

3.6. Circulating Tumor DNA (ctDNA) and Immunotherapy

In general, ctDNA could be used for mutation analysis GIST, but the relevance and usefulness specifically in setting of immunotherapy in GIST unclear. Consider to delete this paragraph.

Conclusion

The review provides a clear overview of clinical trials with immunotherapies in GIST. However, the transition from chapter 3 to conclusion is very abrupt. It would be very valuable to provide a summary of result of clinical trials (positive and negative results) with a clear opinion of authors concerning position of immunotherapy in GIST.

The TKI’s have gained a clear position with high effectiveness in GIST with relatively good tolerance and acceptable financial costs.  As immunotherapy is a potential therapeutic target for GIST, its value in addition to TKI’s, should be discussed in terms of additional effectiveness, adverse events, patient’s burden and financial costs.

Author Response

1. Please include in the first the main GIST epidemiological; the incidence, the primary sites of occurrence, age, and gender distribution, for the reader to have a better context of the disease.

We incorporated the epidemiology, incidence, primary sites, age, and gender distribution of GIST.

…affecting 1.1 every 100,000 persons per year worldwide. While GISTs can occur throughout the GI tract, 60% of the cases present in the stomach and around 30% in the small intestine. GISTs can also occur at any age but are most common in the 40-60s age group with similar male to female incidence.

2. Lines (43-44): please specify clinical benefit (progression free survival, overall survival) to emphasize the dramatically improved prognosis of patients with advanced GIST after treatment with imatinib and eventually to provide the reader a reference for future novel therapies

We added the PFS and OS of patients treated with imatinib.

“leading to an impressive median overall survival (OS) of 3.9 years, and progression free survival (PFS) of 1.9 years”

3. Paragraph 2: unclear purpose of this paragraph. Please elaborate.

The original idea of the paragraph was to explain that M2 macrophages have an immunosuppressive nature and are more abundant in advanced diseases. Somewhat this idea is already conveyed in the next paragraph, so we decided to omit entirely the second paragraph.

4. Paragraph 3 (lines 93-94): please provide reference for statement on decreased ration of CD8+/T cells and reduced HLA class 1 in GIST.

We provided the reference.

5. Line 102: please define the meaning of acronym PI.

We defined in the text the meaning of PI. It stands for proliferation indexed, and in the original study was measured by Ki67 antigen.

6. Line 131-132: although not common, neutropenia (and rarely agranulocytosis) induced by imatinib, has been observed in GIST patients. Please mention this observation with reference (case report, PMID: 25810235).

Thank you for pointing such consideration, we certainly will incorporate it.

“The preservation of the proper maturation of the leukocytes makes immunotherapy an attractive adjuvant option in GIST treatment, since imatinib, except in rare cases, doesn’t lead to leukopenia.”

7. Please check the list of relevant trials for completeness.

Consider to add following trials with immunotherapy in GIST in table 1.

Regorafenib with Avelumab (NCT03475953)

Pembrolizumab in combination with cyclophosphamide in advanced sarcomas (NCT02406781)

Axitinib in combination with avelumab (NCT04258956)

We included in the table the abovementioned trials.

8. Please mention whether the patient, in the clinical trial (imatinib + peg-intreferon alfa-2b) were pre-treated with imatinib or any other TKI’s. And what was duration of response?

All of the 8 enrolled patients were treatment naïve, with no previous imatinib or TKI. We clarified this in the text.

The follow-up with the patient with a complete response was 3.6 years, and the original article did not report any relapse/PD. Only two out of the 8 patients had PD, one at 2.1yr and the other at 2.2yr. The authors of the original article did not calculate the median PFS of the patients.

“Regarding outcomes, all eight patients had partial responses, with one of the patients having a complete pathological response. Out of 8 patients, only two had PD disease during the study’s follow up (median follow up of 3.4 years). The patient with a CR, had no PD during his 3.6 years follow up.”

9. Lines 239-244: Only 3 patients (of 33 patients) were GIST patients in this trial (VEGF inhibitor + pembrolizumab). If the goal of the review is to include all the studies performed with immunotherapy in GIST patients, more studies should be mentioned for completeness (including trials with only few GIST patient). However, in my opinion only relevant trials with adequate numbers of GIST patients should be described. Therefore describing of this trial could be skipped.

We agree with your assessment, and therefore will focus on trials with an adequate number of GIST patients. We omitted such a trial.

10.It’s doubtful whether therapy with anti-KIT antibodies can be defined as immunotherapy. Consider to delete this paragraph.

Some monoclonocal antibodies can also be considered a type of immunotherapy if they interact with the immune system:

https://www.cancer.gov/about-cancer/treatment/types/immunotherapy#what-are-the-types-of-immunotherapy

I this case, we opted to report about anti-KIT antibodies, as the proposed mechanism by which this monoclonal antibody destroys GIST cells, is thru the increase of immunosurveillance (PMID: 23382202). We followed the discussion with Abrams et al work, to put the first findings into perspective, since when they did not conjugate a kit-antibody with a drug, it did not destroy the GIST cells. We changed the wording to make it clearer the immune aspect of the therapy.

“Monoclonal antibodies can also enhance immune cell-mediated tumor clearance. SR1 is an anti-KIT monoclonal antibody that can slow the growth of human GIST cells in vitro and in GIST mice. After treatment with the antibody, tumor cells showed decrease in KIT expression, leading the authors to consider that the mechanism by which the antibodies affect the cells is and downregulation of KIT, and through augmenting phagocytosis of GIST cells by macrophages. “

11. In general, ctDNA could be used for mutation analysis GIST, but the relevance and usefulness specifically in setting of immunotherapy in GIST unclear. Consider to delete this paragraph.

Although we believe that ctDNA could play a future role in patient selection, currently it is still not fully incorporated in immunotherapies in GIST. Thus, we decided to omit such a paragraph.

12. The review provides a clear overview of clinical trials with immunotherapies in GIST. However, the transition from chapter 3 to conclusion is very abrupt. It would be very valuable to provide a summary of result of clinical trials (positive and negative results) with a clear opinion of authors concerning position of immunotherapy in GIST.

We appreciate the suggestion, and we incorporated a new discussion session to address it. Our position is that currently, immunotherapies could serve in a synergic role with current TKIs.

13.The TKI’s have gained a clear position with high effectiveness in GIST with relatively good tolerance and acceptable financial costs. As immunotherapy is a potential therapeutic target for GIST, its value in addition to TKI’s, should be discussed in terms of additional effectiveness, adverse events, patient’s burden and financial costs.

In the new discussion session, we address the effectiveness, and limitations of immunotherapies

Reviewer 2 Report

This is a well written review on immunotherapy strategies in GIST, a topic of current and growing interest. 

However, for this reason many paper have been already recently published about it, in order to improve the review, I suggest to enrich the section on the immunological background of GIST, adding several refecences missing. In particular, what has been recently found about the differences in immunological background of GIST according to mutational status is completely missing and deserve to be reported. Moreover, in addition with data on tumor-infiltrating immune cells reported in GIST, also data on immunoprofiling and high-affinity neoepitopes production accordind to mutational status should reported. 

The paragraph on ctDNA and immunotherapy, included into the immunotherapies in GIST section, is not appropriated for that section. The authors could discuss it as future perspective into conclusion section. 

In table 1, some ongoing clinical trials are missing: NCT03475953 (REGOIMMUNE); NCT04258956 (AXAGIST); NCT02834013.

A table with published resulats on immunotherapy strategies should be informative.

Author Response

1. I suggest to enrich the section on the immunological background of GIST, adding several refecences missing. In particular, what has been recently found about the differences in immunological background of GIST according to mutational status is completely missing and deserve to be reported.

We have expanded the discussion of the immunological background of GIST with new references.

2. Moreover, in addition with data on tumor-infiltrating immune cells reported in GIST, also data on immunoprofiling and high-affinity neoepitopes production according to mutational status should reported.

We have included the discussion of neo-epitopes as per the mutational status.

3. The paragraph on ctDNA and immunotherapy, included into the immunotherapies in GIST section, is not appropriated for that section. The authors could discuss it as future perspective into conclusion section.

As ctDNA is still not fully incorporated in immunotherapies in GIST, we decided to omit such a paragraph.

 4. In table 1, some ongoing clinical trials are missing: NCT03475953 (REGOIMMUNE); NCT04258956 (AXAGIST); NCT02834013.

Thank you for the suggestion. We readily included in the table the abovementioned trials.

5. A table with published resulats on immunotherapy strategies should be informative.

We included the results of the concluded trials in our table. 

Reviewer 3 Report

Arshad et al. has summarized immunotherapy research revolving GIST. The review is well written and structured.

Author Response

Thank you for the positive feedback.